# Predicting Oligomeric states of Fluorescent Proteins using Mamba

Agney K Rajeev[1], Joel Joseph K B[1], and Subhankar Mishra[*1]

[1]National Institute of Science Education and Research (an OCC of HBNI), Bhubaneswar, India - 752050
{agneyk.rajeev,joeljoseph.kb,smishra}@niser.ac.in

## Abstract

Fluorescent proteins (FPs) are essential tools in biomedical imaging, known for their ability to absorb and emit light, thereby allowing visualization of biological processes. Understanding the oligomeric state is crucial, as monomeric forms are often preferred in applications to minimize potential artifacts and prevent interference with cellular functions. Experimental methods to find the oligomeric state can be time-consuming and expensive. Most of the current computational model is CPU-based, limiting their speed and scalability. This paper studies the effectiveness of GPU-based deep-learning models in predicting the oligomeric states of fluorescent proteins directly from their amino acid sequences, specifically focusing on the Mamba architecture. Various protein-specific augmentations were also employed to enhance the model's generalizability. Our results indicate that the mamba-based model achieves accuracy and F1 score close to 90% and an MCC value of 0.8 with in predicting the oligomeric states of fluorescent proteins directly from its amino acid sequence. The code used in this study is available at GitHub repository.

## 1 Introduction

Fluorescent proteins are essential tools in biomedical imaging, derived from marine organisms like the jellyfish *Aequorea victoria*. They absorb light at one wavelength and emit it at another, enabling various applications such as imaging and tracking in biological research.

In order to effectively utilize fluorescent proteins, it is important to understand their oligomeric state, whether they exist as monomers or oligomers. Protein oligomerization is the process by which two or more protein monomers non-covalently bind together to form a larger, multimeric complex known as an oligomer. These oligomers can range from dimers to larger structures, and their formation can affect the protein's functional properties, stability, and interactions. The oligomeric state can significantly influence a fluorescent protein's behavior and suitability for specific applications. Monomeric fluorescent proteins are often preferred for fusion constructs to avoid potential artifacts caused by protein-protein interactions that can occur with oligomers. For instance, research has shown that the tetrameric structure of DsRed can lead to complications; for example, Mizuno et al[1] demonstrated that DsRed aggregation disrupts the normal function of calmodulin in the cytosol. Similarly, Zacharias[2] highlighted that the oligomerization of fluorescent proteins can interfere with the signaling pathways of target proteins when these proteins are used as tagging probes in fluorescence resonance energy transfer (FRET) experiments.

This necessitates the development of computational models to predict the oligomeric state of fluorescent proteins directly from their protein sequences. Such predictive capabilities could help in selecting or engineering fluorescent proteins with desired oligomeric properties. Experimentally, researchers rely on in vitro techniques like gel filtration, ultracentrifugation, and electrophoresis or in vivo techniques like yeast-2-hybrid screens and FRET[3]. All these experimental techniques consume significant time, cost, experimental resources, and expertise in protein biology. Given these challenges in finding the oligomeric state analytically, data-driven approaches like machine learning have gained notable interest. However, the lack of sufficient data and standard benchmarks poses significant challenges in developing robust predictive models and benchmarking them.

In this study, we propose a machine learning algorithm based on the Mamba architecture to predict the oligomeric state directly from protein sequences. Mamba has been used across various domains, including vision[4], video[5] medical image classification[6]. Our contributions include: (1) This paper presents the first application of the Mamba architecture to protein classification, demonstrating its effectiveness in addressing this problem. (2) The model is built on the Mamba architecture, enabling scalability using powerful GPUs, which overcomes the limitations of previous models that predominantly relied on CPU-based decision trees. (3) Achieved superior performance in sensitivity analyses compared to RNN and LSTM models while also reducing computational costs and utilizing fewer parameters. (4) Achieved accuracy of 89.97% and an F1 score of 89.28% in ten-fold cross-validation on a dataset cu-

---

*Corresponding Author.

Proceedings of the 6th Northern Lights Deep Learning Conference (NLDL), PMLR 265, 2025.

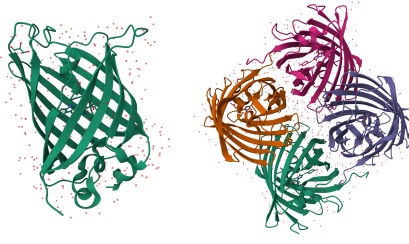

**Figure 1.** Structure of NowGFP[8] and DsRed[9] as an example for monomeric and tetrameric protein respectively

rated by Simeon et al. [7], performing better than the model developed by Simeon et al. and all other known deep learning models for this specific problem, to the best of our knowledge.

## 2 Literature Review

Garian (2001)[10] was the first to predict protein oligomeric structure directly from its amino acid sequence. The author has used a decision tree classifier (C4.5) and, demonstrated that primary sequences do contain quaternary structure information, achieving an accuracy rate of 70% in distinguishing between homodimers and non-homodimers.

Subsequent research expanded on this idea. Zhang et al. (2003)[11] used an SVM algorithm for distinguishing between homodimers and non-homodimers. Song (2007)[12] applied a nearest-neighbor algorithm to discriminate between homo-oligomers. Chou and Cai (2003)[13] used pseudo amino acid composition, developed by Chou (2001)[14], to predict the oligomeric state, improving predictive capabilities by preserving sequence order. Carugo (2007)[15] predicted the tendency of protein chains to form hetero-oligomers with 80% accuracy. Xiao et al. (2010)[16] introduced "Quat-2L," a web server using a fuzzy k-nearest neighbor algorithm to predict the oligomeric state. Quat-2L's two-layer model classified proteins as monomers, homomers, or heteromers with a success rate of 71.14%, and further categorized homo-oligomers and hetero-oligomers into subtypes with success rates of 76.91% and 82.52%, respectively.

Simeon et al. (2016)[7] introduced "osFP," a web server for predicting the oligomeric states of fluorescent proteins (FPs) from amino acid sequences. The authors curated a comprehensive dataset of 409 FP sequences from the literature, each annotated with its oligomeric state. Using the J48 decision tree algorithm, they achieved accuracy, sensitivity, and specificity over 80%, with a Matthews Correlation Coefficient (MCC) over 0.6. The authors hosted their predictive model on a web server named "osFP." which allows users to input the amino acid sequences of FPs and receive predictions regarding their oligomeric state.

Agarwal et al (2020)[17] present a novel approach to predicting the oligomeric states of fluorescent proteins (FPs) using artificial neural networks, specifically autoencoders. The authors utilized the same dataset curated by Simeon et al. They applied autoencoders to compress the amino acid sequence data, transforming 5237 nodes into a more manageable set of 32 nodes. This compression aimed to retain essential features necessary for predicting oligomeric states. A decision tree was then incorporated to enhance the predictive power of the autoencoder by identifying significant amino acids influencing the oligomeric state. This decision tree contributed 8 more nodes to already existing 32 nodes from the encoder, resulting in a total of 40 nodes to encode a protein. This encoded protein is then passed into a neural network consisting of 4 dense layer and a gaussian noise layer for the prediction task. The combined approach achieved a prediction accuracy of 70%. Since Agarwal et al (2020) did not specify their evaluation method, it is not possible to directly compare the performance of our model with theirs, especially considering the differences in test-train splits and whether they used single-run values or cross-validations. In the end, the study underscores the broader application of deep learning models in predicting the oligomeric state of fluorescent proteins.

Tam and Zhang (2021)[18] developed FPredX, a machine learning-based tool for predicting key properties of FPs from amino acid sequences. The study addresses the challenge of accurately predicting FP properties such as excitation and emission maxima, brightness, and oligomeric states, which are critical for various applications in bioimaging and molecular biology. FPredX utilizes gradient-boosted decision tree models trained on a dataset curated from FPbase. They benchmarked their approach against the dataset curated by Simeon et al. (2016)[7]. FPredX achieved superior results, with precision, recall, and F1 scores of 91.3%, 95.5%, and 93.3%, respectively, performing better than the "osFP" model, which had 84.1%, 87.9%, and 85.9% in the same evaluation.

## 3 Methods

In this section, we outline the methods and techniques employed for our study.

### 3.1 Models

We employ five models for our study.

The **k-Nearest Neighbour (kNN)** algorithm uses pairwise sequence similarity as a distance metric and classifies a sequence as the label that occurs most often among the labels of the k sequences

most similar to it according to pairwise similarity. This procedure is motivated by the Basic Local Alignment Search Tool (BLAST), which is widely used in biotechnology and bioinformatics.

The S6 state space model **Mamba**[19] has emerged as a promising architecture for sequence modeling due to their linear scaling with sequence length. Inspired from control engineering, the Structured State Space Sequence Model(S4) uses the mathematical model of state-space representation constituting a set of input, output, and variables connected by first-order differential equations. Mamba built on the foundations of S4 and uses a time-varying framework to selectively focus on relevant information within sequences.

State space models define a sequence-to-sequence transformations in two stages.

$$
\begin{aligned}
h'(t) &= Ah(t) + Bx(t) \\
y(t) &= Ch(t)
\end{aligned}
\tag{1}
$$

We define a MambaBlock according to the architecture proposed by Dao and Gu using a State Space Model (SSM) layer [19] which combines an H3 layer mentioned in [20] with a gated MLP followed by a root mean square normalization layer (RMSNorm[21]) as shown in fig. 2. Within the block, we use a convolution layer of kernel size 3, linear projection layers with a forward expansion of 2 and SiLU(Sigmoid-Weighted Linear Units) activations[22].

We also experiment with simpler, closely related models, **RNN** and **LSTM**, as well as the state-of-the-art architecture **Transformer (Tfmr)** model used frequently in protein classifications to provide competitive baselines for comparison.

## 3.2 Data Augmentation

In an attempt to improve the generalizability of the model and improve the size of the dataset, we apply various protein-specific augmentations to our dataset mentioned by Shen et al[23] and Sun et al[24]. We chose protein-specific augmentations because traditional methods of augmentations can change the functional aspect of the protein and may affect the model's performance negatively. The augmentations employed on amino acid sequence $S = \{S_i\}$ were:

**Random Substitution(A1)**: With a probability of 10%, we randomly substitute an amino acid with the amino acid most similar to it to maintain its properties. The substitution mappings used were those optimized by Shen et al.

**Sequence Reversion(A2)**: The amino acid sequence is reversed to read from the C-terminal to the N-terminal instead of the usual N-terminal to C-terminal. This changes the protein structurally and functionally but could improve the model's recognition of local features.

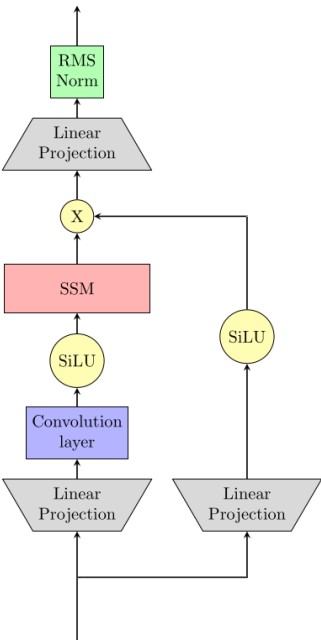

**Figure 2.** Architecture of the MambaBlock (X represents a multiplication between inputs)

**Subsequence Sampling(A3)**: Randomly select an amino acid from the sequence and form a subsequence along with the 49 amino acids following it(if the chosen amino acid is less than 49 amino acids away from the final amino acid, then select all amino acids till the final amino acid). Treat the subsequence as the augmented sequence. This is another method to try and encourage the model to recognize local features.

**Subsequence Shuffling(A4)**: Select a subsequence from the sequence by a similar procedure to A3 but of length 20. Shuffle the amino acids of this subsequence and replace them in the initial sequence. This promotes the recognition of properties such as amino acid counts, which do not depend on the order of the amino acids in a sequence.

**Random Swap(A5)**: Each amino acid in the sequence exchanges its position with another amino acid in the sequence with a probability of 10%.

**Random Deletion(A6)**: Each amino acid in the sequence is deleted with a probability of 10%.

## 4 Experiments

### 4.1 Datasets

We mainly use the dataset curated by Simeon et al. containing 409 fluorescent protein sequences separated into Monomers, Dimers, and Oligomers, which will be referred to as the OSFP dataset in further discussions. We combine Dimers and Oligomers of the original dataset as one class of Oligomers. Thus, our final dataset consists of 409 protein sequences

separated as Monomers and Oligomers. The dataset is fairly balanced, containing 207 monomers and 202 oligomers.

Additionally, we attempted to reproduce several test results using a separate dataset, which was independently curated from the community-edited open-source fluorescent protein database, FPBase[25]. We extracted 656 proteins with annotated oligomerization states out of which 381 were monomers (labeled as 'Monomer') and the remaining 275 were either dimers or tetramers (labeled as 'Oligomer')

## 4.2 Implementation

The k-Nearest Neighbors (kNN) model was implemented utilizing pairwise sequence similarity from the Biopython library as the distance metric.

The RNN, LSTM and Transformer models are implemented using the RNN, LSTM and TransformerEncoder API respectively of the nn module of PyTorch Library.

The Mamba model was implemented from the open-sourced codebase provided by Gu et al.

For the deep learning models, the sequences are tokenized and padded with each amino acid as a single token, resulting in a total of 21 tokens (20 amino acid tokens + 1 padding token). The tokenized sequences are fed into an embedding layer, which converts each amino acid into a vector representation with 512 components. Each vector representation passes through the models. To reduce overfitting, a dropout layer with a 30% rate is applied. Afterward, a dense layer is used to generate a 2-dimensional output vector. All hyperparameters used in the model were finalized after optimization. The input-output behaviour of our models are highlighted in figure. 3

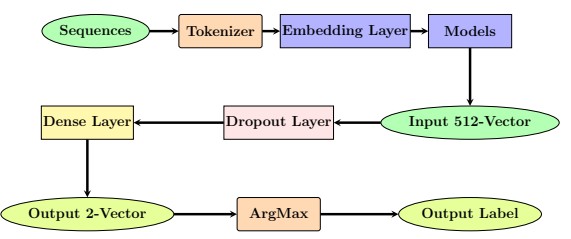

**Figure 3.** Schematic representation of the input-output behaviours of our models. The argmax function here takes in a vector as input and outputs the label corresponding to the highest valued component.

## 4.3 Experimental setup

Two testing methods used in the study are:

1. **10-fold Cross-Validation test(10CV)** where the dataset is partitioned into 10 near equal subsets. The models are tested on one subset

after training on the remaining 9 subsets. This is repeated for all subsets and averaged.

2. **Independent dataset test(ID)** where 70 data points are randomly selected from the whole dataset on which the model will be tested after training on the remaining data points. This process is repeated and averaged over 100 random seeds.

The models were evaluated on the metrics of Accuracy, F1 score(F1) and Matthews Correlation Coefficient(MCC).

MCC is evaluated by equation.2

$$MCC = \frac{TP \times TN - FP \times FN}{(TP+FP)(TP+FN)(TN+FP)(TN+FN)} \tag{2}$$

The MCC metric is found to handle data imbalances better than both Accuracy and F1 score as well as provide symmetrical importance to both positive and negative predictions.The metrics we employed were to obtain direct comparisons with Simeon et al (2016).

Sensitivity to hyperparameters such as state size for Mamba, Hidden size for RNN, LSTM, and Transformer, and number of blocks in Mamba were also studied. The computation cost in GMACs (1GMAC= $10^9$MACs) and total Parameter Count for each model were also monitored throughout.

A MAC(Multiply-Accumulate) operation constitutes computing a multiplication of two floating point numbers followed by an addition of the product to an accumulator. During the forward pass, MAC operations are used to compute the weighted sums in neurons, which are then passed through activation functions and during backpropagation, MAC operation is involving in computing gradients. A singular MAC operation is given by the equation 3 where a,b and c are floating point variables.

$$a \leftarrow a + (b \times c) \tag{3}$$

All experiments were executed on a Linux server equipped with Intel(R) Xeon(R) Gold 6138 CPU @ 2.00GHz, 512 GB RAM, and 4 NVIDIA GeForce RTX 2080 Ti with 11GB VRAM each.

## 5 Results

The results for the tests mentioned in 3 are provided. First, we compare the effects of data augmentations with the baseline Mamba model on the 10CV test. Then, we compare the results of all the models with the best augmentation from the previous results in 10CV and the independent dataset, along with their computation cost and parameter size. A misclassification of 1 sequence results in a 0.24% decrease in

accuracy on 10x cross-validation, and a misclassification of 1 sequence results in a 0.014% decrease in accuracy on a 100x averaged independent dataset evaluation. Finally, we try to optimize the hyperparameters to find the model with the lowest computation cost without compromising on performance metrics.

## 5.1 Data augmentations

The effects of various data augmentation techniques provided in sec.3.2 are documented in table.1. All six mentioned augmentations were applied individually and the three augmentations with an accuracy greater than or equal to the baseline model were combined and tested. A combination of random substitution and sequence reversion proved to be the most effective, performs better than other augmentations in all metrics, and achieves an accuracy close to 90% and an MCC greater than 0.8. The ID test results for the kNN, baseline Mamba as well as Mamba running on the dataset augmented by random substitution and sequence reversion are provided in table 2.

## 5.2 Benchmarking the performance

The models were benchmarked with 10CV test and ID test with the dataset provided by Simeon et al[7]. The models were also benchmarked in self curated dataset from FPbase through 10CV. All the results are provided in tables 3, 4, 6.

## 5.3 Sensitivity tests

In order to find a set of optimum hyperparameters to minimize the computational cost and parameter count while still maintaining high-performance metrics, hyperparameters such as state size for Mamba and Transformer, Hidden size for RNN and LSTM were also studied and are given in Fig. 4. Smallest Mamba model achieves performance of an accuracy of about 88.5% with a computation cost of 2.15 GMACs and a Parameter count of 578562 which is significantly lower than other models with similar performance metrics.

Additionally, we compared the runtime of our deep learning model(Mamba) with the currently available CPU-based models by Simeon et al.(osFP) and Lambert et al.(FPredX). Sample sequences of length from 12 to 500 amino acids were curated from the Uniprot Reviewed database[26] as FASTA files and runtime were compared using the hardware mention in 4.3. Results are presented in table.7

| Augmentation | Accuracy | F1 | MCC |
|---|---|---|---|
| Baseline | 0.8875 | 0.8831 | 0.7760 |
| A1 | 0.8899 | 0.8845 | 0.7775 |
| A2 | 0.8875 | 0.8851 | 0.7721 |
| A3 | 0.8801 | 0.8757 | 0.7623 |
| A4 | 0.8851 | 0.8823 | 0.7714 |
| A5 | 0.8825 | 0.8772 | 0.7648 |
| A6 | 0.8899 | 0.8863 | 0.7784 |
| **A1+A2** | **0.8997** | **0.8928** | **0.8001** |
| A2+A6 | 0.8801 | 0.8723 | 0.7567 |
| A1+A6 | 0.8851 | 0.8769 | 0.7672 |
| A1+A2+A6 | 0.8900 | 0.8832 | 0.7798 |

**Table 1.** Comparison between various augmentations and combinations of augmentations for the 10CV test

| Model | Accuracy | F1 | MCC |
|---|---|---|---|
| kNN | 0.8571 | 0.8572 | 0.7131 |
| Baseline | 0.8751 | 0.874 | 0.7527 |
| **A1+A2** | **0.8770** | **0.8759** | **0.7570** |

**Table 2.** Comparison kNN, Baseline Mamba, and the best augmented Mamba on the ID test

| Model | Accuracy | F1 | MCC |
|---|---|---|---|
| RNN | 0.8748 ± 0.0432 | 0.8673 ± 0.0909 | 0.7531 ± 0.0909 |
| LSTM | 0.8938 ± 0.0443 | 0.8913 ± 0.0601 | 0.7892 ± 0.0601 |
| Tfmr | 0.8082 ± 0.0663 | 0.8065 ± 0.0984 | 0.6280 ± 0.1308 |
| Mamba | 0.8768 ± 0.0267 | 0.8785 ± 0.0745 | 0.7608 ± 0.0596 |

**Table 3.** Comparison of RNN, LSTM Transformer and Mamba on the 10CV test on OSFP

| Model | Accuracy | F1 | MCC |
|---|---|---|---|
| RNN | 0.7794 ± 0.0756 | 0.6278 ± 0.2214 | 0.4429 ± 0.2214 |
| LSTM | 0.7713 ± 0.0688 | 0.6197 ± 0.2082 | 0.4201 ± 0.2049 |
| Tfmr | 0.7352 ± 0.0978 | 0.5558 ± 0.1888 | 0.3741 ± 0.1998 |
| Mamba | 0.7720 ± 0.0807 | 0.6353 ± 0.2386 | 0.4197 ± 0.1926 |

**Table 4.** Comparison of RNN, LSTM Transformer and Mamba on the 10CV test on the FPBase dataset

| Model | Accuracy | F1 | MCC |
|---|---|---|---|
| RNN | 0.8589 ± 0.0406 | 0.8557 ± 0.0804 | 0.7223 ± 0.0804 |
| LSTM | 0.8775 ± 0.0388 | 0.8754 ± 0.0811 | 0.7593 ± 0.0747 |
| Tfmr | 0.7901 ± 0.0463 | 0.7861 ± 0.0945 | 0.5892 ± 0.0912 |
| Mamba | 0.8781 ± 0.0389 | 0.8768 ± 0.0734 | 0.7590 ± 0.0734 |

**Table 5.** Comparison of RNN, Transformer and Mamba on the ID test on OSFP dataset

## 6 Conclusion

In this study, we have developed a Mamba-based model to predict the oligomeric state of fluorescent protein directly from its amino acid sequence. The ability to distinguish if a fluorescent protein is a monomer or an oligomer is important for biologists to design novel fluorescent proteins with a desired

| Model | Accuracy | F1 | MCC |
|---|---|---|---|
| RNN | 0.8289 ± 0.0455 | 0.7951 ± 0.0883 | 0.6625 ± 0.0883 |
| LSTM | 0.8387 ± 0.0443 | 0.8063 ± 0.0874 | 0.6806 ± 0.0874 |
| Tfmr | 0.7391 ± 0.0548 | 0.6825 ± 0.0911 | 0.5011 ± 0.1019 |
| Mamba | 0.8560 ± 0.0423 | 0.8260 ± 0.0869 | 0.7104 ± 0.0869 |

**Table 6.** Comparison of RNN, Transformer and Mamba on the ID test on FPBase dataset

| No of Sequences | OSFP | FPredX | Mamba |
|---|---|---|---|
| 10 | 0.03 ± 0.03 | 38.48 ± 0.58 | 1.90 ± 0.04 |
| 100 | 0.06 ± 0.00 | 39.69 ± 0.48 | 1.53 ± 0.02 |
| 1000 | 0.48 ± 0.01 | 55.82 ± 0.49 | 2.07 ± 0.02 |
| 10000 | 4.80 ± 0.16 | 472.72 ± 1.70 | 13.31 ± 0.12 |
| 100000 | 54.66 ± 4.24 | 61989.49±2.01 | 126.09 ± 0.20 |

**Table 7.** Comparison of Runtime of CPU-based models and Mamba model

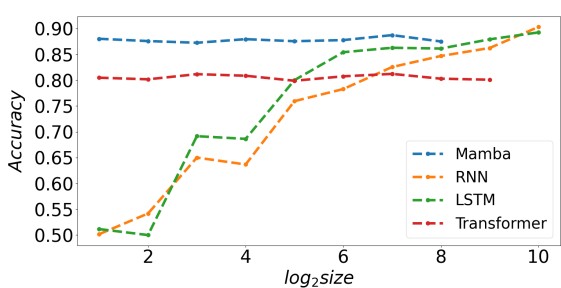

**Figure 4.** log plot of accuracy sensitivity of all models with hidden/state size

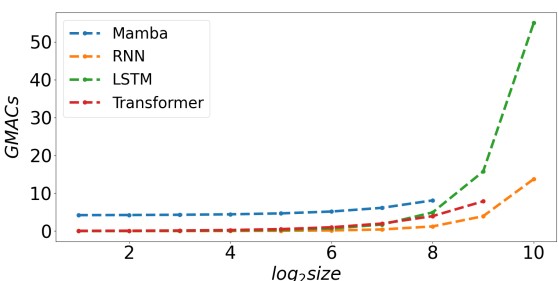

**Figure 5.** log plot of GMACs of all models with hidden/state size

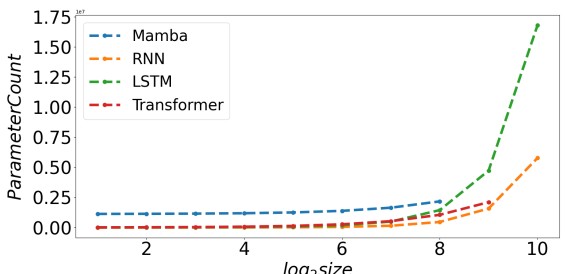

**Figure 6.** log plot of Parameter Count (in units of $10^7$) of all models with hidden/state size

oligomeric state, as a fluorescent protein's oligomeric state is crucial for bio-medical imaging and FRET experiments. Research has shown oligomeric pro-

| Hyperparameter | Optimal Value |
|---|---|
| batch_size | 8 |
| n_layers | 2 |
| dim_model | 512 |
| hidden_size(RNN,LSTM) | 1024 |
| hidden_size(Tfmr,Mamba) | 128 |
| n_head(Tfmr) | 8 |
| d_conv(Mamba) | 3 |

**Table 8.** Optimized hyperparameters for all models

teins tend to form unwanted artifacts that need to be avoided for proper experimental results.

We observe that our Mamba model performs better than all other models in terms of performance metrics while still maintaining the lowest computation cost and parameter count. Mamba demonstrates a significantly better runtime compared to the FPredX model by Lambert et al.[25] while also achieving superior performance metrics than those reported for the model by Simeon et al.[7], albeit with a slightly longer runtime. Regarding data augmentations, a combination of random substitution and sequence reversion augmentation techniques provided the best results on the 10CV test. This may be because random substitution with amino acids of same class would help to preserve the functional characteristics and sequence reversal would encourage the model to learn more short-range relations thus improving the model's generalizability. The data augmentation techniques did not provide any drastic improvements on the ID test but nevertheless improved the model to achieve an MCC > 0.8 and an accuracy and F1 score close to 90% in the 10CV test.

Additionally, considering the linear scaling of the Mamba model w.r.t sequence length, our model could perform well on the prediction of oligomeric states and other properties of general protein sequences. Furthermore, new and improved proteinspecific augmentations could be experimented on the dataset to possibly enhance the results further.

# Acknowledgments

We would like to express our gratitude to Sagar Prakash Barad for their invaluable insights and suggestions that greatly contributed to the development of the ideas presented in this article.

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

# A   Data Augmentation Algorithms

---

**Algorithm A.1** Random Substitution(A1)

---

**procedure** RANDOMSUBSTITUTION($S$)
    $map \leftarrow \{A : V, S : T, F : Y, K : R, C : M, D : E, N : Q, V : I\}$
    $n \leftarrow len(S)$
    Initialize empty sequence $S'$ of length $n$
    **for** $S_i$ in $S$ **do**
        **if** randint(0,100) < 10 and $S_i$ in $map$ **then**
            $S'_i \leftarrow map[S_i]$
        **else**
            $S'_i \leftarrow S_i$
        **end if**
    **end for**
    **return** $S'$
**end procedure**

---

---

**Algorithm A.2** Sequence Reversion(A2)

---

**procedure** SEQUENCEREVERSION($S$)
    $n \leftarrow len(S)$
    Initialize empty sequence $S'$ of length $n$
    **for** $S_i$ in $S$ **do** $S'_{n+1-i} = S_i$
    **end for**
    **return** $S'$
**end procedure**

---

---

**Algorithm A.3** Subsequence Sampling(A3)

---

**procedure** SUBSEQUENCESAMPLING($S$)
    $n \leftarrow len(S)$
    $r \leftarrow$ randint(1,n)
    $S' \leftarrow S[r : min(r + 50, n)]$
    **return** $S'$
**end procedure**

---

**Algorithm A.4** Subsequence Shuffle(A4)

---

**procedure** SUBSEQUENCESHUFFLING($S$)
    $n \leftarrow len(S)$
    $r \leftarrow$ randint(1,n)
    $s \leftarrow S[r : min(r + 50, n)]$
    $S' \leftarrow S[: r] + shuffle(s) + S[min(r + 50, n) :]$
    **return** $S'$
**end procedure**

---

**Algorithm A.5** Random Swap(A5)

---

**procedure** RANDOMSWAP($S$)
    $n \leftarrow len(S)$
    Initialize empty sequence $S'$ of length $n$
    **for** $S_i$ in $S$ **do**
        **if** randint(0,100) $< 10$ **then**
            $j \leftarrow$ randint(1,n)
            $S'_i \leftarrow S_j$
            $S'_j \leftarrow S_i$
        **else**
            $S'_i \leftarrow S_i$
        **end if**
    **end for**
    **return** $S'$
**end procedure**

---

**Algorithm A.6** Random Deletion(A6)

---

**procedure** RANDOMDELETION($S$)
    $n \leftarrow len(S)$
    Initialize empty sequence $S'$
    **for** $S_i$ in $S$ **do**
        **if** randint(0,100) $\geq 10$ **then**
            $S' \leftarrow S' + S_i$
        **end if**
    **end for**
    **return** $S'$
**end procedure**

---

