# OpenReview forum: "Predicting Oligomeric states of Fluorescent Proteins using Mamba"
_NLDL.org/2025/Conference — NLDL 2025 Oral_

### Official Review · Reviewer_yi5B · 2024-10-09
**Predicting Oligomeric states of Fluorescent Proteins using Mamba**

**Confidence:** 5

**Summary:**

This paper test the new Mamba architecture to predict (classify) whether a fluorescent protein's structure will be monomeric or oligomeric based on the amino acid sequence. Two (standard) datasets are used, created and curated by others. The main experiments are run on a dataset (FP) of totally 409 proteins and then to some extent repeated on a separate set of 656 proteins (OSFP set). Some augmentation approaches are compared and the overall performance is compared to other methods/networks previously used for this. The results show slightly higher numbers (translated to ~1 more sequence correctly classified) than previously reported but the model is computationally more effective.

**Strengths:**

It is good and important to evaluate and modify new approaches to different applications. It would be good with more details on the application/biological use. Is this an example of protein structure (3D and quarternary) prediction in general? How many fluorescent proteins are there where this would be interesting to use? Who would be a user- the companies selling fluorescent proteins, others? Did it lead to any biological insights, or how could the results be further analyzed to get biological insights?

**Weaknesses:**

*Figure 1, copy from another publication- copyright?
*Figure 1- caption is a bit confusing. DsRed is the right figure (tetramer). Also, one cannot say "monomeric oligomerization"

*Explain what the MCC is and why you use it (not standard).

*All parts and abbreviations in the model should be (briefly at least) explained - what is H3 and give ref and/or explain SiLu.

*It is confusing what you call your datasets - the "name" or abbreviation you use should be given when they are described.

*Unclear why you do 10-fold cross-validation AND your ID approach. What do you wish to see/understand from the two approaches?

*Is there really a difference to the baseline for any of the augmentation methods? What is the difference between 0.8875 and eg. 0.8899 in your experimental setup?

*Why don't you show the variance/spread between your repeats (10 and 100 respectively)?

*I lack a proper exploration or more thorough investigation of the results. Which sequences are problematic? Can any biological and/or patterns be identified?

*Same Q also for the other results. How do the numbers relate to your dataset sizes? what do they actually mean? How many more are actually correctly classified? To me it seems like the claim "outperforms" is rather strong for these numbers.....

*Papers on arxiv that are published in journals/conferences should be referenced to the journal/conference.

**Justification:**

This is a typical student project: Try this new model on a dataset and see how it performs. The novelty is very low and so is the significance (very low on biology non-existing on technology). The presentation also lacks in clarity.

The only reason for a non-reject decision is that it is important to encourage students to "test" research approaches and practice on writing scientific papers. The presentation and motivation of the paper can be improved with rather little effort but the very low novelty and significance will not change.

---

> ### Author Rebuttal · Authors · 2024-10-22
>
> We thank the reviewer for their valuable feedback.
>
> **>Response to protein structure prediction in general**
>
> Thank you for your feedback. Our model does not predict the quaternary structure of proteins; rather, it identifies the most likely label that the fluorescent protein may assume. We have expanded paragraph 2 in the Introduction section, explaining its application.
>
> **>Response to interesting use cases in fluorescent proteins**
>
> There are numerous fluorescent proteins whose oligomericity is not yet known. As an example, NanoLuc in FPbase [1] has its oligomericity listed as unknown. Our in silico approach could also benefit the development of de novo fluorescent proteins. However, it is challenging to provide a precise number, as this area is continually evolving with ongoing research.
>
> [1] T. J. Lambert. “FPbase: a community-editable fluorescent protein database”. In: Nature Methods 16.4 (Mar. 2019), pp. 277–278. doi:10.1038/s41592-019-0352-8. url: https: //www.nature.com/articles/s41592-019-0352-8.
>
> **>Response to 'Who would be a user?'**
>
> Thank you for your thoughtful question. Understanding the oligomericity of a fluorescent protein is crucial for its biological applications. This information would be of immense interest to companies involved in the development of novel de novo proteins, as our method provides a fast, efficient, and accurate in silico alternative to time-consuming and costly wet lab experiments.
>
> **>Response to issues with Figure 1**
>
> Thank you for bringing this to our attention. We had previously cited the source in the main text, but we have now added the citation to the figure caption as well. Additionally, we have revised the caption for clarity, including correcting the reference to DsRed as the tetramer and adjusting the language regarding "monomeric oligomerization."
>
> **>Response to the explanation of MCC**
>
> Thank you for your comment. MCC stands for Matthew’s Correlation Coefficient, defined as follows;
>
> \begin{equation}
> MCC =\frac{TP \times TN - FP \times FN}{(TP + FP)(TP + FN)(TN + FP)(TN + FN)}
> \end{equation}
>
> The MCC metric is found to handle data imbalances better than both Accuracy and F1 score as well as provide symmetrical importance to both positive and negative predictions. The metrics we employed were to obtain direct comparisons with Simeon et al (2016). We have also added the same to the paper in Section 4.3
>
> **>Response to the addition of references to H3 and SiLU**
>
> Thank you for your comment. The necessary references are added.
>
> **>Response to confusing abbreviations for dataset names**
>
>  Thank you for your feedback. We have addressed this issue by ensuring that the names or abbreviations of our datasets are provided when they are described in the paper.
>
> **>Response to use of 10-fold cross-validation and Independent Dataset approach**
>
> We used the same dataset and validation methods followed by Simeon et al (2016) to draw a direct comparison. Simeon et al.’s dataset is frequently used to construct and benchmark similar models in predicting fluorescent protein oligomeric states in various papers.
>
> **>Response to the difference in performance of augmentation methods**
>
> Thank you for your comment. Although we agree that the difference is not so significant in the individual augments, however, that helped us in search of the combination of augmentation methods that gave us significant improvements over the baseline as shown in Table 1.
>
> **>Response to lack of spread between the repeats**
>
> Thank you for your comment. We have updated our table with standard deviations.
>
> **> Response to the identification of biological patterns**
>
> Thank you for your comment. Currently, this is out of scope for this work, however we certainly would like to appreciate the comment, and we would consider adding this in our future work.
>
> **>Response to the relation of numbers to the dataset sizes**
>
> Thank you for the comment. A misclassification of 1 sequence results in a 0.24% decrease in accuracy on 10x cross-validation, and misclassification of 1 sequence results in a 0.014% decrease in accuracy on a 100x averaged independent dataset evaluation. The word ‘outperforms’ has been replaced with ‘performs better than’ across the paper.
>
> **>Response to updating arXiv references**
>
> We have updated the references to ensure that papers on arXiv that have been published in journals or conferences are properly cited in their respective venues.
>
> Please refer to the revised manuscript for updates.

---

### Official Review · Reviewer_KfRX · 2024-10-09
**A neural network approach to predict state of fluorescnet proteins such as GFP and dsRed.**

**Confidence:** 3

**Summary:**

Fluorophores used in biomedical imaging may consist of a single fluorescing unit, or a cluster of several units, influencing their ability to provide a useful signal. This paper investigates the possibility to predict if proteins exist as single or grouped units, based on the sequence of the protein.
This field of applying deep neural networks is a bit far from my expertise, but I have done my best in asses strengths and weaknesses. First of all, the strengths include approaches for improving on the output from a tool called Mamba, by having large focus on reducing computational time by moving from a slower CPU-based approach to faster and more efficient GPU processing. The number of papers in the field is limited, but the authors claim they outperform a previous studies utilizing models such as RNN and LSTM.

**Strengths:**

On the weakness-side is the very limited explanation of the field of protein 'clumping', making it tricky for the reader to understand input as well as output from the proposed models. The discussion becomes even harder to follow for someone with limited understanding of the application area.  Improving on the output from a tool called Mamba, by having large focus on reducing computational time by moving from a slower CPU-based approach to faster and more efficient GPU processing, can potentially be useful aslo for other approaches to understand moleccular mechanisms. In general, I believe the paper is sound and well-structured.

**Weaknesses:**

Due to its slightly limited novelty and lack of information to convey the purpose of the project, I will rank it lower.

**Final Rebuttal Confidence:**

3

**Final Rebuttal Justification:**

Using deep learning methods to categorize proteins as forming aggregates or not based on their amino acid sequence is a rather narrow field, and likely interesting for a limited audience. The major drawback of the paper is the lack of comparison with previous developed methods in the field (other reviewers point to FPredX); however, the authors plan to do this comparison prior to submitting their final  photo-ready version, according to the reply to @m2uG. Apart from these two concerns, the paper is now, with some revisions. very well written and nicely structured. I therefore recommend a weak accept.

**Justification:**

I justify my positive assessment by the fact that I believe a broad application in the biomedical and cell biology field will benefit from this type of publications. The full assessment of the method is difficult to judge with my limited knowledge in the application field, but strategies for training, validation and testing seem to be sound, and the input data is balanced.

---

> ### Author Rebuttal · Authors · 2024-10-22
>
> We thank the reviewer for their valuable feedback.
>
> **>Response to the explanation for protein oligomerization**
>
> Thank you for your comments. In response, we have added additional details explaining protein oligomerization to provide clarity on this topic in paragraph 2 of the Introduction.
> The input-output behavior of our model has also been expanded upon and is now included in the implementation section (4.2) of the paper along with a schematic.
>
> Please refer to the revised manuscript to see the updates.

---

### Official Review · Reviewer_hT39 · 2024-10-14
**Review of the paper "Predicting Oligomeric states of Fluorescent Proteins using Mamba"**

**Confidence:** 4

**Summary:**

This paper presents a method for categorizing the oligomeric states of proteins based on their amino acid sequences. The novelty of the research lies in its use of deep learning frameworks, particularly the recently developed Mamba architecture. The authors demonstrates that this approach outperforms other deep learning methods, such as LSTM, RNN, and Transformer models, in terms of accuracy, sensitivity, and computational efficiency.

**Strengths:**

The primary objective of this paper appears to be improving computational efficiency in predicting oligomeric states, with a secondary focus on increasing sensitivity in categorization. While the justification for using the Mamba architecture is not clearly articulated, it seems that Mamba has been tested and validated in DNA modeling, a field closely related to protein sequencing. Given that Mamba is a newly emerged framework, testing it's perfomance towards other deep learning architectures represents a valuable contribution to current research on Mamba's potential use cases. The paper thoroughly explores the use of various data augmentation techniques, including ablation studies. It is well-written and adheres to academic standards, introducing the reader to the topic of oligomeric state determination in a clear and concise manner. Additionally, the inclusion of a reference to the codebase used for testing enhances the paper’s transparency and provides deeper insights into its technical aspects.

**Weaknesses:**

The literature review mentions the Gradient Boost decision tree-based method FPredX, which reportedly achieves F1 scores of 93.3%, significantly higher than the results presented in this paper. Why is this not addressed in the conclusion? A decision tree approach would also offer superior explainability, a topic that is notably absent from the discussion. While lower computational costs (GPU vs. CPU-based, as mentioned) are advantageous, should this come at the expense of accuracy and sensitivity? Additionally, no direct metric comparison between FPredX and this method is provided, so can we be certain this approach performs better in terms of computational time? Given the brief format of this conference paper, I recommend shortening the somewhat lengthy literature review and focusing more on discussing these critical aspects.

The paper compares performance to other deep learning frameworks, such as RNN, LSTM, and Transformer, but does not specify which implementations are used. Upon reviewing the code, it appears that standard PyTorch API calls are employed for testing these, but this should be explicitly mentioned in the text.

While the paper claims that hyperparameter optimization was performed, it does not detail the method used. Was grid search employed? From what I gather in section 5.3, the smallest Mamba model achieves the best performance... Furthermore, the sensitivity tests in section 5.3 combine various augmentations to evaluate their effects, but only 11 combinations are tested, despite there being 6 different techniques, which could result in 64 possible combinations. The rationale for selecting these 11 baselines is not explained.

Additionally, the graphs in Figures 3-5 show that the Mamba parameter size stops at 10^8, several orders of magnitude smaller than those of the other methods. Why?

Finally, there are a few minor grammatical issues. Ensure that all acronyms, such as MCC and MAC, are defined. In line 206, the sentence '...overfitting of data, following which a dense layer...' could benefit from rewording. Similarly, in line 264, the phrase '...attempted to reproduce a few test results...' needs clarification for greater precision. Also, in line 147, there is a punctuation mark following a reference, while the sentence continues. Please review for clarity and correctness.

**Final Rebuttal Confidence:**

3

**Final Rebuttal Justification:**

Based on the revision and comments from the authours in the process of rebuttal, and the discussions among the reviewers as followed, I belive the paper can be presented at NLDL 2025.

**Justification:**

The paper regrettably demonstrates no comparison towards the stated methods previously developed by research in this field, forming a major drawback. As such, the paper needs to develop these baselines, preferably before being presented at the conference. Also, some minor flaws were highlighted according to the need for detailing more on experimental setup. However, gaining broader knowledge on the application Mamba as a novel, rather unexplored deep learning framework is advantageous for the community. Moreover, Mamba seems justifiable for protein sequencing, where processing time can be a critical factor, even if the evidence for this was lacking.

---

> ### Author Rebuttal · Authors · 2024-10-22
>
> We thank the reviewer for their valuable feedback.
>
> **> Response to comparison with FPredX**
>
> Thank you for your insightful comments. Regarding FPredX, a direct comparison is currently challenging due to differences in testing and validation methodologies, as well as distinct training datasets used in their model versus ours. The Gradient Boosting Decision Tree model, especially due to its CPU implementations, often struggles with the scalability required for large-scale applications, whereas our model demonstrates superior scalability.
>
> **>Response to lack of implementation details**
>
> Thank you for the feedback. We have created a new implementation section(4.2) outlining the necessary details
>
> **>Response to lack of all possible augmentation techniques**
>
> Thank you for your comment. Upon analysis of each individual augmentation, we observed that only augmentations A1, A2 and A6 proved to not be detrimental to the performance in comparison to the baseline. Thus, only the combinations of the mentioned augmentations were considered for further analysis.
>
> **>Response to parameter size of Mamba**
>
> Thank you for the comment. The implementation of the Mamba model we used limits the state size parameter to $2^8$.
>
> Please refer to the revised manuscript to see the updates.

---

### Official Review · Reviewer_m2uG · 2024-10-15
**Promising Approach with Mamba Architecture for Predicting Protein Oligomeric States, but Lacks Key Comparisons and Clarity**

**Confidence:** 4

**Summary:**

This paper proposes a method to predict the oligomeric state of fluorescent proteins using a deep learning model based on the Mamba architecture. The authors apply different data augmentation methods to enhance the model’s generalizability and compare it to other sequence-based models like RNN, LSTM, and Transformer. The Mamba-based model achieves high accuracy while reducing computational costs compared to other models.

**Strengths:**

The paper contributes Mamba architecture scheme for predicting the oligomeric state of fluorescent proteins, which improves accuracy and computational efficiency compared to traditional models. The introduction of protein-specific data augmentation techniques enhances the model's generalizability, offering a new direction for handling limited datasets in protein classification tasks.

**Weaknesses:**

The introduction lacks relevant background on the Mamba architecture and its prior applications, as well as the role of data augmentation in this context. Expanding on these topics would provide a clearer foundation for understanding the paper's contributions.

While the paper highlights the advantages of GPU-based methods, it would benefit from additional experiments comparing the Mamba-based model with CPU-based approaches discussed in the literature review section. This comparison could further emphasize the performance improvements, and the literature review could be condensed to make room for these experiments.

In section 3.1, the model implementation details could be moved to a separate "Implementation" section, making the paper more organized.

Table 2 is not explicitly referenced in the main body, which diminishes its usefulness.

Lastly, the paper does not discuss specific sequence patterns or features in a protein's amino acid sequence that may influence its oligomeric state. Addressing this could provide deeper biological insights and strengthen the interpretability of the model's predictions.

**Justification:**

The paper presents a promising method using the Mamba architecture to predict the oligomeric state of fluorescent proteins, improving computational efficiency and accuracy compared to traditional models. Its use of data augmentation techniques is a strong contribution.

However, the paper lacks sufficient background on Mamba and data augmentation, which are crucial to understanding the approach. Additionally, the absence of direct comparisons with CPU-based methods weakens the evidence for the model's improvements. Organizational issues, such as missing references to key tables, also detract from clarity. While the contributions are notable, these limitations reduce the overall impact of the work.

---

> ### Author Rebuttal · Authors · 2024-10-22
>
> We thank the reviewer for their valuable feedback.
>
> **>Response to relevant background**
>
> Thank you for your insightful comment. We have updated Section 3.1 by providing additional details about the Mamba architecture. Additionally, we have revised the Introduction to include a discussion on the applications of Mamba, citing the relevant papers. The data augmentations described in Section 3.2 were applied to enhance the model's generalizability, ultimately improving its performance metrics.
>
> **>Response to the suggestion of comparison to CPU-based methods**
>
> Thank you for your valuable feedback. We appreciate the suggestion of including a comparison with CPU-based approaches. We are currently working on including the CPU-based approach $osFP$, and we anticipate having the results ready by the camera-ready submission. We will include them in the final version of the paper.
>
> **>Response to the addition of the Implementation section**
>
> Thank you for the feedback. We have added section 4.2 titled “Implementation” detailing the model implementations.
>
> **>Response to explicit reference to Table.2**
>
> We thank the reviewer for pointing out this, We have added the reference to Table 2 in Section 5.1.
>
> **>Response to discussion of specific sequence patterns or features in a protein's amino acid sequence**
>
> Thank you for your insightful comment. While the exploration of specific sequence patterns or features influencing a protein's oligomeric state is beyond the current scope of this work, we greatly appreciate your suggestion. This is an important aspect, and we will certainly consider incorporating such analyses in our future research to provide deeper biological insights and enhance model interpretability.
>
> Please refer to the revised manuscript to see the updates.

---

### Meta-Review · Area_Chair_4qBe · 2024-11-02

**Recommendation:** Accept (Oral)
**Confidence:** 4

**Metareview:**

The paper proposes a method using the Mamba deep learning architecture to predict the oligomeric state (monomeric vs. oligomeric) of fluorescent proteins based on their amino acid sequences. The model is tested for accuracy, sensitivity, and computational efficiency against traditional methods, including RNN, LSTM, and Transformer models, as well as gradient-boosted decision trees (FPredX), as well as CPU-based models ( in an updated version of the manuscript). Various data augmentation methods are also applied to enhance generalizability.

Strengths:
1) The proposed work effectively applies the Mamba architecture to predict oligomeric states of fluorescent proteins. It achieves high accuracy while reducing computational costs compared to traditional CPU-based methods.
2) Applying Mamba ( a newly emerged framework) to protein sequence analysis offers a new direction in handling limited datasets in protein classification tasks.
3) By shifting from CPU to GPU processing, the model has shown efficiency improvements that could be relevant to various biomedical and molecular research applications.
4) To improve the model's generalizability for limited datasets, the authors have also introduced some novel protein-specific data augmentation techniques, particularly useful in protein classification tasks.
5) Testing Mamba's potential beyond DNA modeling provides new insights, and including a reference to the codebase enhances transparency.
6) The work offers the potential for broader biological application. For example, the proposed work has the potential to contribute insights into protein structure prediction and fluorescent protein use.
7)The manuscript's current version is well-written, well-structured, and meets academic standards.

Weaknesses:
1) Reviewers discuss that the novelty and significance of the work, particularly in terms of biological insights and technical advancement, need some improvements and make the paper potentially interesting to a limited audience.
2) The first version of the manuscript lacks a direct comparison with FPredX and CPU-based methods. However, the authors have addressed these issues in the revised version of the model.
3) The introduction in the first version of the manuscript needs more background/ references on the Mamba architecture, and specific terms (e.g., MCC, SiLu) and dataset details need to be clarified. However, the authors have addressed these issues in the revised manuscript.
4) The first version of the manuscript contains some methodological and structural clarity issues, as well as some issues regarding presentation quality. For example, references for certain figures, more details on hyperparameter optimization, and experimental results are needed. However, these issues have been improved in the revised manuscript.

In sum, in the revised version of the manuscript, the authors have addressed most reviewer concerns, particularly the lack of comparison with previously developed methods in the field, especially FPredX and CPU-based methods. Moreover, the current version of the work is well-written and meets academic standards. Given the authors' efforts to address the major concerns of reviewers and generally positive consensus among the reviewers after revision (one acceptance, two-week acceptance, and one weak rejection), it seems reasonable to lean toward recommending acceptance and oral presentation.

**Suggested Changes To The Recommendation:**

2: I'm certain of the recommendation.  It should not be changed

---

### Decision · Program_Chairs · 2024-11-06

**Decision:**

Accept (Oral)

**Comment:**

We recommend an oral and a poster presentation given the AC and reviewers recommendations.